# Integrative Transcriptomic and Metabolomic Analyses of the Mechanism of Anthocyanin Accumulation and Fruit Coloring in Three Blueberry Varieties of Different Colors

Liwei Chu [†] , Qianhui Du [†], Aizhen Li, Guiting Liu, Hexin Wang, Qingqing Cui, Zhichao Liu, Haixia Liu, Yani Lu, Yanqiong Deng and Guohui Xu *

College of Life and Health, Dalian University, No. 10 Xuefu Street, Dalian 116622, China; chuliwei@dlu.edu.cn (L.C.); duqianhui@s.dlu.edu.cn (Q.D.); liaizhen@s.dlu.edu.cn (A.L.); liuguiting@s.dlu.edu.cn (G.L.); whexin@sina.com (H.W.); cuiqq_caas@163.com (Q.C.); liuzhichao@s.dlu.edu.cn (Z.L.); liuhaixia@s.dlu.edu.cn (H.L.); 18702410280@163.com (Y.L.); dyq18822119272@163.com (Y.D.)
* Correspondence: xugh520@163.com
† These authors contributed equally to this work.

**Abstract:** Blueberries are recognized worldwide as one of the most important healthy foods due to their anthocyanins, which have special antioxidant properties. They have become a highly produced and valuable fruit crop. Most blueberry varieties are rich in anthocyanins, which impart a beautiful blue color; however, there are currently several blueberry varieties with different colors worldwide, and these special-colored varieties are the key to analyzing the coloring mechanism of blueberry fruit. Fruit color could be seen as an important nutritional quality trait in terms of marketing. In this study, a combination of transcriptomic and metabolomic analyses was performed on three representative blueberry varieties ('Pink Popcorn', 'Chandler', and 'Black Pearl') with pink, blue, and black fruits, respectively. The metabolomic results showed that the delphinium pigment is the dominant anthocyanin, which is the prerequisite for the formation of fruit color in blueberries. We identified 18 candidate structural genes in the anthocyanin biosynthesis pathway that were significantly up-regulated during three stages of fruit ripening in 'Black Pearl' and 'Chandler', but these were not found to be significantly expressed in 'Pink Popcorn' after combining the transcriptomic analysis results. The non-expression of the *VcANS* gene may lead to the pink color of the mature fruit of 'Pink Popcorn'. The phylogenetic tree, heatmap analysis, and WGCNA analysis identified a candidate transcription factor, *VcMYBA,* which may regulate the differences between black and blue fruits in blueberries by regulating the expression level of multiple structural genes in the anthocyanin biosynthesis pathway. These results provide new insights into the mechanisms of anthocyanin accumulation and coloration in blueberries during fruit ripening and can help support production practices to improve fruit quality characteristics. The key candidate genes that regulate the fruit color differences among different blueberry varieties have the potential to enhance the antioxidant properties and quality characteristics of blueberries through future genomic editing.

**Keywords:** anthocyanin biosynthesis; blueberry; fruit color; transcriptome; metabolome; fruit ripening





## 1. Introduction

Blueberry (*Vaccinium* spp.) is a perennial deciduous or evergreen low shrub native to North America [1]. Blueberry fruit tastes sweet and is rich in organic acids, phenols, minerals, and vitamins. Previous studies have shown that the nutritional components of blueberries exhibit significant antioxidant, anti-inflammatory, and anti-cancer activities. The active components in blueberries can help slow aging in humans, improve vision, enhance memory, and reduce the risk of heart disease [2–9]. Blueberries ranked first among the 15 healthy foods listed by an authoritative British nutritionist and were listed as one

of the five healthy foods by the International Food and Agriculture Organization [10]. Currently, they are widely cultivated in Asia, Europe, South America, Africa, Australia, and New Zealand [11,12]. Previous studies have shown that anthocyanins are the main active substances in blueberries that can improve human health [13]. The anthocyanin content can be as high as 4.95 mg/g in blueberries; so, they have long been regarded as one of the most abundant plant sources of anthocyanins [14].

Anthocyanins are water-soluble pigments present within plants that have important functions in plant physiology as well as possible beneficial health effects. Several studies have shown that anthocyanins and polyphenols can prevent chronic diseases in humans [4–6]. Anthocyanins can prevent human cardiovascular disease and diabetes due to their anti-inflammatory and antioxidant activities [7–9]. Anthocyanins, which have nutritional benefits and attract frugivores to help spread seeds, determine the degree of blueberry fruit coloration [15,16]. During the ripening process of blueberry fruit, fruit color greatly affects fruit quality and the accumulation of healthy chemicals [17].

The anthocyanin biosynthesis pathway (ABP) is a branch of secondary metabolites containing flavonoids. Initially, the structural genes involved in flavonoid biosynthesis were identified and collated in model plants such as *Arabidopsis thaliana*, *Zea mays*, and *Petunia hybrida* [18]. Although the composition and accumulation patterns of anthocyanins vary greatly in different species, the main reactions involved in anthocyanin synthesis are basically the same. It is believed that anthocyanin synthesis is a conserved process that is triggered and initiated by a series of related enzyme actions [19,20]. Initially synthesized via phenylalanine deamination, dihydroflavonol is catalyzed by *CHI*, *CHS*, *F3H,* and other enzymes in the early stage; anthocyanin (unstable state) is subsequently synthesized through the actions of *F3′H*, *F3′5′H*, *DFR*, *ANS,* and other enzymes and is transformed into a stable-state anthocyanin under the action of glucose transferase *UFGT*. Finally, it is transported by glutathione S-transferase (GST) to the vacuole for storage [21–23]. Anthocyanins are eventually produced through more than 20 steps, accumulating during the plant fruit ripening process, which is related to the gradual deepening in fruit color [24].

In addition to these structural genes in ABP, it has been found that the transcription factor MYB or MBW (composed of MYB, bHLH, and WD40) complex could regulate the expression levels of these structural genes in anthocyanin synthesis [25–28]. MYB transcription factors, widely identified and cloned in plants, were found to play a regulatory role in controlling various functions of plant metabolism, development, signal transduction, and biotic/abiotic stresses [29,30]. R2R3-MYB-like transcription factors have been widely reported to be involved in the regulation of anthocyanin synthesis [31]. A total of 126 R2R3-MYB-like transcription factors were identified in *Arabidopsis* [32]. *AtMYB75/AtPAP1* and *AtMYB90/AtPAP2* in *Arabidopsis thaliana*, *VvMYBA1* and *VvMYBA2* in *Vitis vinifera*, and *MdMYB1* in apple were found to specifically contribute to anthocyanin biosynthesis by regulating *UFGT* and *DFR* expression. The overexpression of *AtMYB75/90/113/114* in *Arabidopsis thaliana* causes pigmentation in anthocyanin biosynthesis in leaves and seedlings [33–38]. Many studies have also shown that multiple MYB and bHLH transcription factors play an inhibitory role in ABP [39]. Blueberries are known for being nutrient-rich and for their unique flavor. Anthocyanins are pivotal nutrients in blueberries. The synthesis of anthocyanins is a key step in the formation of color traits during the fruit ripening process. Therefore, the cellular and molecular mechanisms of coloration during blueberry fruit ripening have attracted much attention from scientists in recent years [40]. Delphinidin, cyanidin, peonidin, petunidin, and malvidin were found to be the main components of anthocyanin glycosides in blueberries [41]. The publication of multiple high-quality blueberry genomes in recent years laid a solid foundation for the study of the molecular biology of blueberries [42–46]. Two expression sequence tag libraries (ESTs) from ripening blueberry fruit were constructed as a resource for flavonoid-related gene identification and expression profile mapping [47]. The expression patterns of 31 enzymes involved in anthocyanin biosynthesis have been identified. Among them, two candidate genes are *CUFF.20951* (*UFGT*) and *CUFF.43605* (*LDOX* or *ANS*), which may control the

high anthocyanin proportions in ripe berry fruit, were highly expressed in red and ripe fruit compared with other genes in this pathway [48]. Multiple studies have shown that the expression levels of *DFR* and *ANS* dramatically increase during the mid-ripening of blueberry [47,49,50]. The genetic variations of blueberry anthocyanin were found to be associated with genes related to the glycosylation, acylation, and methylation of the molecule [51–53].

*VcMYBA* is a regulator of anthocyanin synthesis that strongly activates the *DFR* promoter of blueberries. It may be the central activator of berry skin pigmentation [54]. Metabolomics and transcriptomes were combined to analyze the dynamic changes in the flavonoid synthesis of the fruit ripening of the northern high bush blueberry variety 'Nui' and rabbit eye blueberry variety 'Velluto Blue'. Additionally, other transcriptional activators, such as *MYBPA1*, *bHLH2*, and inhibitory factor *MYBC2*, which are involved in the regulation of blueberry anthocyanin biosynthesis, were identified [17]. The novel co-regulation of *MYBA*, *MYBPA1*, and *MYBPA2* with synthase genes in anthocyanin biosynthesis was revealed in *Vaccinium myrtillus* [55]. The stable overexpression of *VcMYBA1* in blueberries elevated anthocyanin content in transgenic plants [56]. In this experiment, eleven MYBs, seven bHLHs, and six WD40s in blueberry fruit were identified via homologous sequence analysis using the amino acid sequences of the homologous MYB-bHLH-WD40 complex in *Arabidopsis thaliana*, apple, grape, and strawberry. Further verification of *VcMYBL1*, *VcbHLHL1*, and *VcWD40L2*, which are involved in the regulation of ABP during blueberry fruit maturation and color development, was performed [57].

However, the understanding of the ABP regulatory mechanisms in blueberries is still incomplete. The color of blueberries gradually darkens during ripening. Typically, blueberry fruit is green and of varying size during the initial "swelling" stage of fruit development before accumulating a red pigment at the beginning of ripening, which continuously tints to blue as the fruit grows and finally becomes dark blue when the fruit reaches maturity [47]. Black is the darkest color in blueberry fruit, indicating a relatively high anthocyanin content. Blue is the most common color in cultivated blueberry varieties. Pink is an extremely rare fruit color mutant in blueberry varieties, containing a very low anthocyanin content. In this study, we performed multiple omics to analyze the differences in coloring mechanisms among these three colors of blueberries. Metabolomics was used to identify and quantify anthocyanins in ripe blueberry fruits of black, blue, and pink color. Transcriptomics was used to identify the key candidate genes involved with the fruit color differences between the 'Black Pearl', 'Chandler', and 'Pink Popcorn' varieties. Additionally, the result was verified by real-time quantitative polymerase chain reaction (qRT-PCR). Phylogenetic and WGCNA analysis was also used to explore blueberry anthocyanin synthesis-related MYBs. The candidate genes identified in this study provide valuable insights for further research on the regulatory network in anthocyanin biosynthesis and will further improve the exploitation of different fruit color *Vaccinium* accessions.

## 2. Materials and Methods

### 2.1. Plant Materials

The black-fruited blueberry variety 'Black Pearl', the blue-fruited blueberry variety 'Chandler', and the pink-fruited blueberry mutant 'Pink Popcorn' were provided and planted in Dalian Senmao Modern Agriculture Co., Ltd., Dalian, China (Blueberry Breeding Cooperative Experimental Base of Dalian University, 121°59′ E, 39°19′ N, altitude: 67 m).

We divided the coloration of blueberry fruit into three stages. Stage 1: The initial fruit appeared greenish in color. Stage 2: The fruit gradually changed from green to red. Stage 3: The fruit is at the ripening stage, where coloring is complete. We collected the fruit of the 'Black Pearl', 'Chandler', and 'Pink Popcorn' varieties at three stages of developmental coloration (a temperature of 25 °C, soil pH = 4.5–5.5, consistent management conditions, and a humidity of 70%). All materials had three biological replicates sampled; each replicate consisted of 10 g samples mixed with more than three individuals, immediately frozen in liquid nitrogen, and then stored at −80 °C for further experiments.

### 2.2. Identification of Pigments

According to a previous study [58], pigmentation was identified in the three stages of fruit development in 'Black Pearl', 'Chandler' and 'Pink Popcorn'. In this study, 50 mg of fruit tissue powder (ground in liquid nitrogen) and 200 uL of methanol were added to a 1.5 mL centrifuge tube. After vertexing, they were mixed very well, and 200 μL of water and chloroform were added, which were then mixed thoroughly and centrifuged at 13,000 rpm using a QuickSpeed 4000 (Monad Biotech Co., Ltd., Jiangsu, China). Mixing was used to obtain the fractionated layer via static settlement. The upper layer consisted of anthocyanins, and the lower layer consisted of carotenoids.

### 2.3. Relative Quantification of Anthocyanin by Ultraviolet–Visible (UV/Vis) Spectroscopy

According to the previous study [59], 15 mL 0.1 mol/L hydrochloric acid was mixed with 1 g fruit tissue that had been ground in liquid nitrogen. Then, the anthocyanins were extracted in a constant-temperature water bath (Guohua Instrument Manufacturing Co., Ltd., Jiangsu, China) at 32 °C for 4 h. The anthocyanin content of each sample was quantitatively measured using an ultraviolet spectrophotometer at a wavelength of 530 nm. The absorbance of 0.1 mol/L hydrochloric acid at 530 nm was used as a blank control. The absorbance reading at 530 nm for each 0.1 mol/L sample was used as the unit of measurement for the relative anthocyanin content. Graphing was performed using GraphPad 7.00 software.

### 2.4. Anthocyanins Identification by Metabolite Profiling

For the identification and quantification of anthocyanins from the 'Black Pearl', 'Chandler', and 'Pink Popcorn' fruits at Stage 3, anthocyanin content was detected using MetWare (Metware Biotechnology Co., Ltd., Wuhan, China) based on the AB Sciex QTRAP 6500 LC-MS/MS platform. The sample was freeze-dried and ground (30 Hz, 1.5 min), and then 50 mg of the powder was extracted with 0.5 mL methanol/water/hydrochloric acid (500:500:1, V/V/V). Then, the extract was vortexed for 5 min and subjected to ultrasound for 5 min using a SCIENTZ-IID (Ningbo Scienz Bio-tech Co., Ltd., Ningbo, China) and then centrifuged at 12,000 $g$ for 3 min under 4 °C. The supernatants were collected and filtrated through a membrane filter (0.22 μm, ANPEL Laboratory Technologies Inc., Shanghai, China) before LC-MS/MS analysis. The sample extracts were analyzed using a UPLC-ESI-MS/MS system (UPLC: ExionLC™ AD; MS: Applied Biosystems 6500 Triple Quadrupole, https://sciex.com.cn/ (accessed on 5 June 2023), Shanghai AB Sciex Analytical Instrument Trading Co., Ltd., Shanghai, China). Anthocyanins were analyzed using scheduled multiple reaction monitoring (MRM). Data acquisitions were performed using Analyst 1.6.3 software (Sciex). Multiquant 3.0.3 software (Sciex) was used to quantify all metabolites.

### 2.5. Total RNA Extraction and Quality Assessment

All freeze-dried samples were ground into powder in liquid nitrogen, and 50 mg of the powder was used for total RNA extraction with an RNA extraction kit (Beijing Adelaide Biotechnology Co., Ltd., Beijing, China). RNA concentration, RIN value, 28 S/18 S and fragment size were examined using an Agilent 2100 Bioanalyzer (Beijing, China) to determine the integrity of the RNA. A UV spectrophotometer, NanoDrop™ (Thermo Fisher Scientific Inc., Shanghai, China), was used to test RNA purity and concentration (OD260/280).

### 2.6. Illumina Transcriptome (RNA-Seq) Library Preparation, Sequencing, and Expression Level Estimation

The total RNA of the three blueberry varieties at three coloring stages was extracted for RNA-Seq to reveal the transcriptome expression profile of blueberry fruit coloration during ripening. Three biological replicates were performed at each stage for each variety,

with a total of 27 samples (each replicate consisted of 5 g samples mixed with more than three individuals).

The DNBSEQ platform was used for library construction and quality control. After the samples passed the test, the library was constructed according to the following steps: Enrich mRNA with magnetic beads with Oligo(dT) and add fragmentation buffer to interrupt mRNA. The first cDNA strand was synthesized with six-base random hexamers using mRNA as a template, then the second cDNA strand was synthesized by adding buffer, deoxynucleotide mix (dNTPs), Ribonuclease H (RNaseH), and DNA polymerase I, and then double-stranded cDNA was purified using a kit (Thermos Fisher Scientific Inc., Shanghai, China). The purified double-stranded cDNA was terminal-repaired, and a tail was added with a sequencing joint connected. Finally, PCR amplification was performed to construct the sequencing library. After the library was constructed, the insert range of the library was examined using an Agilent 2100 Bioanalyzer. The ABI StepOnePlus Real-Time PCR System was used to quantify the library concentrations. After passing the quality inspection, sequencing was performed using an Illumina platform sequencer (Beijing, China).

The Internal Perl script was used to process and filter the sequenced raw data to obtain clean reads. HISAT [60] was used to compare the clean reads to the blueberry reference genome [43]. Based on the comparison results, new transcript prediction, differential splicing gene detection, SNP and Indel detection, fusion gene detection, or other analyses were performed. Known and novel genes were quantitatively analyzed, and differential expression analysis was performed based on gene expression levels in different sample groups. The expected maximization (RSEM) method in RNA-Seq was used to estimate gene expression levels by the number of fragments per million mapping readings per thousand base transcripts (FPKM). The resulting *p*-values were adjusted by using Benjamini and Hochberg's method to control the error discovery rate. Differential expression genes (DEGs) were identified with a mutation threshold (FC) of 1.5 and FDR < 0.05. GO function analysis, pathway-function analysis, cluster analysis, protein interaction network, and transcription factor coding ability prediction of the selected DEGs were further explored and analyzed.

### 2.7. Phylogenetic Analysis and Sequence Alignment

The R2R3-MYBS transcription factors of the DEGs sequenced via transcriptome were retrieved from the Blueberry gene bank GDV (www.vaccinium.org, accessed on 5 June 2023) and the sequence was downloaded to retrieve the R2R3-MYBS gene family from the *Arabidopsis thaliana* gene bank TAIR (www.arabidopsis.org, accessed on 5 June 2023) and download its sequence in the fasta format. The R2R3-MYBS gene in blueberry and *Arabidopsis thaliana* was analyzed phylogenetically using MEGA Version 7.0. MUSCLE was employed for sequence alignment, and the Neighbor-Joining method was applied for phylogenetic tree construction. Evologenius (http://www.evolgenius.info/evolview/, accessed on 5 June 2023) was modified for beautification. The blueberry MYB transcription factor was screened with the gene expression profile by using the heatmap analysis method through TBtoolsv2.028.

### 2.8. Co-Expression Network Construction

WGCNA was performed using the WGCNA package in R V4.1.3 [61]. The FPKM expression values of all samples were imported into the WGCNA package, the correlation coefficients between hub genes in the module were calculated, and the co-expression module was established using the automatic network construction function blockwiseModules with deissuesettings, except for the fact that the power was 0.5, the minModuleSize was 50, and mergeCutHeigh was 0.8. All the DEGs were finally clustered into 35 modules. The networks were visualized using Cytoscape v3.9.1.

### 2.9. Quantitative Real-Time PCR Analysis

A total of 26 genes were selected from the DEGs for qRT-PCR to verify the accuracy of the RNA-seq data. The synthesis of cDNA from the total RNA was achieved using a

reverse transcription kit (PrimeScriptTM Reagent Kit, Dalian, China) for qRT-PCR. Every single gene-specific primer design was accessed for *Vaccinium*-BLAST (GDV https://www.vaccinium.org, accessed on 16 May 2023) using the Genome Database and Oligo7. The cDNA template and primer with $2\times$ TB Green™Premix ex Taq™ (Takara, Japan) were fully mixed, and qRT-PCR was performed using a CFX96 Touch Real-Time PCR (Dalian, China) Detection System with a C1000 Touch™ PCR instrument.

The 20 µL reaction mixture of qRT-PCR contains 10 µL of SYBR (Solarbio, Beijing, China), 0.4 µL for each forward and reverse primer with 1 µL of cDNA diluted 10 times in nuclease-free water. The qRT-PCR procedure was set as follows: initial denaturation (95 °C 1 min), followed by 45 PCR cycles (95 °C 10 s, 60 °C 30 s, collection mode: single), melt curve analysis conditions (95 °C 5 s, 60 °C 1 min, 95 °C, collection mode: continuous, collection: 5/°C), and a cooling process (50 °C 30 s). Glyceraldehyde-3-phosphate Dehydrogenase (*GAPDH*) was used as a reference gene. The fluorescence value change curve and solubility curve were obtained at the end of the reaction. The gene expression was calculated as the difference between the Ct values of the target and internal reference gene, referring to the 2-ΔΔCt analysis. The primers were designed by Oligo V7.37 (Supplementary Table S1). The *VcGAPDH1* (maker-VaccDscaff35-augustus-gene-276.34) reference gene of the blueberry was used for normalization.

## 3. Results

### 3.1. Color and Pigment Contents Are Significantly Different during Fruit Ripening of the Three Blueberry Varieties

Tinctorial comparisons were made between 'Black Pearl', 'Chandler', and 'Pink Popcorn' fruits during the three coloring stages. The fruit color of all three blueberry varieties was nearly the same greenish color in Stage 1. In Stage 2, the fruit color of the three blueberry varieties changed from green to red, but the fruit color of 'Pink Popcorn' started to become lighter than that of the other two varieties. At Stage 3, there were significant color differences among the three blueberry varieties (Figure 1). The fruit color of 'Black Pearl' was almost black, and the fruit color of 'Chandler' was dark blue, while the fruit color of 'Pink Popcorn' was still pink.

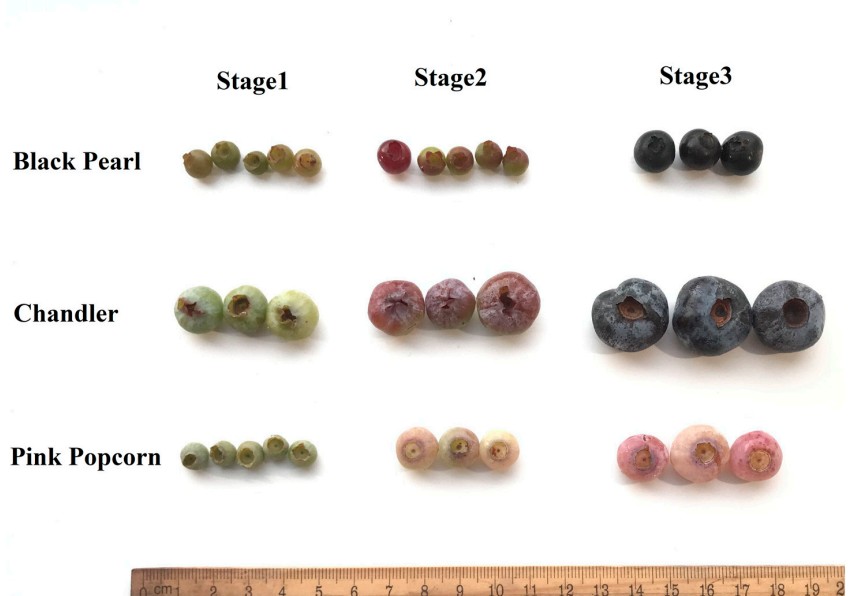

**Figure 1.** Fruit color development during the three stages in the fruit of 'Black Pearl', 'Chandler', and 'Pink Popcorn'.

Pigmentation analysis showed that the color differences in the 'Black Pearl', 'Chandler', and 'Pink Popcorn' varieties were mainly determined by the anthocyanin content

but not carotenoid content (Figure 2a). Anthocyanin accumulation levels increased with coloring at different stages. In Stage 1, there were no significant differences in anthocyanin content among the fruit of three blueberry varieties. Statistical analysis showed significant differences among the anthocyanin content of these three varieties in Stage 2. Compared with Stage 1, the anthocyanin content of 'Black Pearl' and 'Chandler' increased 3–4 times in Stage 2, which was significantly higher than that of 'Pink Popcorn'. At Stage 3, the anthocyanin content of 'Black Pearl' increased ninefold, and that of 'Chandler' increased fourfold compared with Stage 2, while the anthocyanin content of 'Pink Popcorn' fruit remained nearly unchanged (Figure 2b). The accumulation trend of anthocyanin is consistent with the fruit coloring stage of these three blueberry varieties.

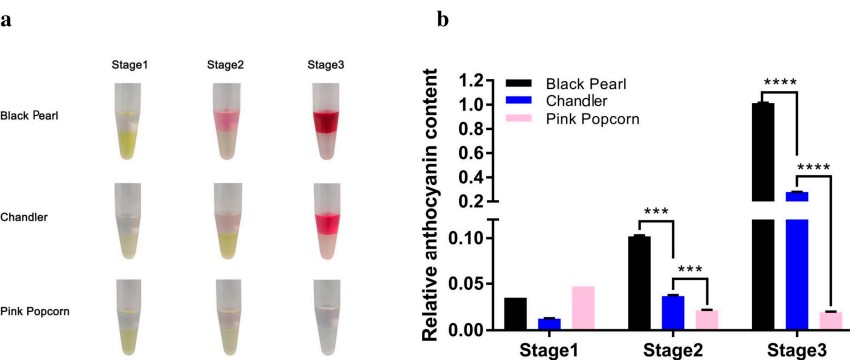

**Figure 2.** Qualitative and quantitative identification of pigment in fruit of 'Black Pearl', 'Chandler' and 'Pink Popcorn'. (**a**) Qualitative identification of pigment in blueberry fruit; (**b**) quantitative identification of pigment in blueberry fruit. *** $p \leq 0.001$, **** $p \leq 0.0001$.

'Black Pearl', 'Chandler', and 'Pink Popcorn' fruits from Stage 3 were collected, and the final anthocyanins in three blueberry fruits were determined via LC-ESI-MS/MS analysis. A total of 12 different anthocyanins were detected, which were divided into two subgroups (Figure 3a,b). Seven delphinidins were detected in both 'Black Pearl' and 'Chandler', and the content of these seven delphinidins in 'Black Pearl' was significantly higher than that in 'Chandler' but were hardly detected in 'Pink Popcorn'. The differences between the 'Black Pearl', 'Chandler', and 'Pink Popcorn' fruits with five cyanidin contents were nearly the same as that of seven delphinidins. In addition, the five proanthocyanidins were all detected in three blueberry varieties (Figure 3c).

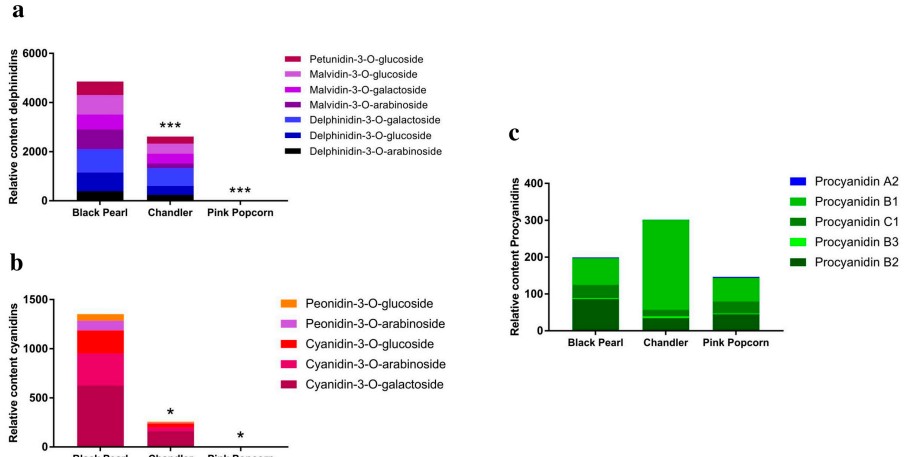

**Figure 3.** Identification and quantification of anthocyanins in fruit of 'Black Pearl', 'Chandler', and 'Pink Popcorn' at last stage of fruit maturity. (**a**) The seven delphinidin-based anthocyanins identified through an LC-ESI-MS/MS system; (**b**) the five cyanidin-based anthocyanins identified with an LC-ESI-MS/MS system; (**c**) the five proanthocyanidins identified using an LC-ESI-MS/MS system; * and *** indicate that there are significant differences detected at $p < 0.05$ or $p < 0.001$.

### 3.2. Key Structural Genes Identified in Anthocyanin Metabolic Pathway through Gene Expression

The fruits of 'Black Pearl', 'Chandler', and 'Pink Popcorn' were sampled at three developmental stages, with three replicates per group, and 27 sample libraries were constructed for RNA-Seq. The average output data per sample was 6.91 Gb. After obtaining clean reads, they were compared to the reference genome using HISAT. An average of 96.04% of reads were mapped to the reference genome with calculated expression levels for each isoform (fragments per kilobase per million, FPKM). After the sequencing reads were aligned to the reference genome and the transcripts had been reconstructed, a total of 90,475 new transcripts were detected, of which 70,229 were new variable splice isoforms of known protein-coding genes, 5917 were transcripts of new protein-coding genes, and the remaining 43,329 were long-stranded non-coding RNAs. The reads were matched to genes by using bowtie2, with an average of 70,593 genes detected per sample. An average of 14,820 DEGs were detected in each group, including 7182 differentially up-regulated genes and 7638 differentially down-regulated genes.

From transcriptome databases, color-transformation-related genes involved in different stages of fruit growth and development in 'Black Pearl', 'Chandler', and 'Pink Popcorn' were screened. According to the FPKM values, the structural genes expression profiles in 'Black Pearl', 'Chandler', and 'Pink Popcorn' during ABP were analyzed through the use of a heatmap, including Chalcone isomerase (*CHI*), Chalcone synthase (*CHS*), Flavanone 3-hydroxylase (*F3H*), Flavonoid 3′-hydroxylase (*F3′H*), Flavonoid3′5′hydroxylase (*F3′5′H*), Dihydroflavonol 4-reductase (*DFR*), Anthocyanidin synthase (*ANS*), and UDP-glucose flavonoid 3-O-glucosyltransferase (*UFGT*).

The expression levels of ABP structural genes in 'Black Pearl' and 'Chandler' increased significantly from Stage 1 to Stage 3, but the expression levels of these genes changed slightly in 'Pink Popcorn' during the three coloring stages (Figure 4a; Supplementary Table S3). In Stage 1, the low expression level of anthocyanin synthesis structural genes in three blueberry varieties conform to the green fruit color. During Stage 2 and Stage 3, the gene expression of 'Black Pearl' and 'Chandler' was significantly increased compared with Stage 1, but there was no significant change in 'Pink Popcorn'. The dynamic expression level difference result was consistent with the fruit color and anthocyanin differences among the three varieties (Figure 1). Compared with 'Black Pearl' and 'Chandler', the expression levels of *VcCHS* and *VcANS* were extremely low in 'Pink Popcorn', but there were slight differences in the expression levels of *VcF3′5′H*.

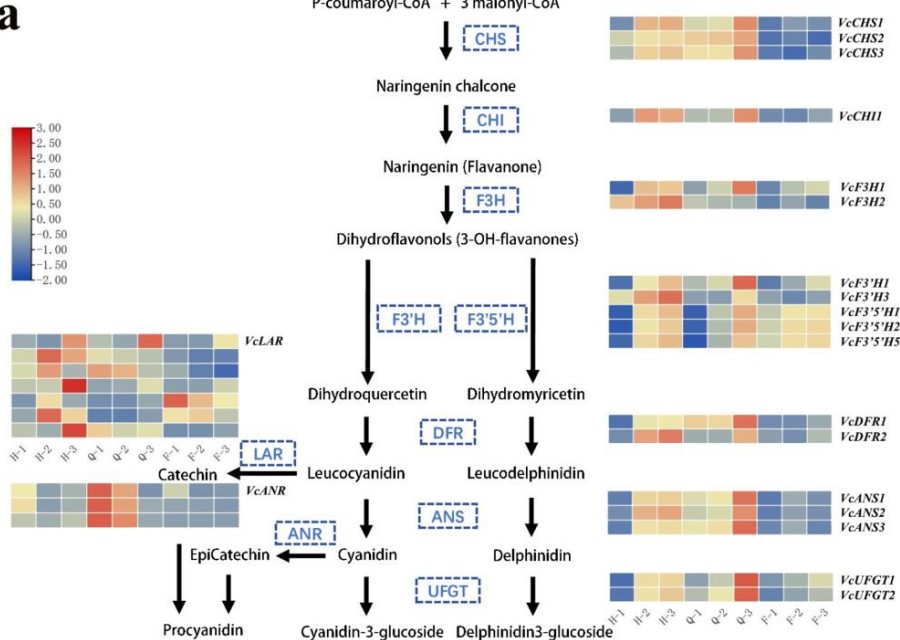

**Figure 4.** *Cont.*

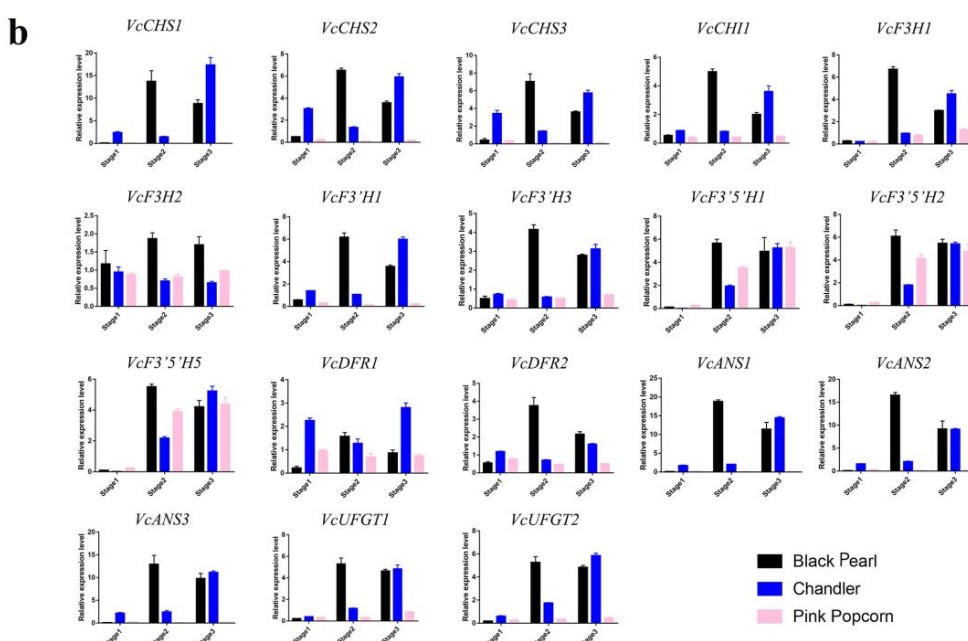

**Figure 4.** Different expression genes in fruit of 'Black Pearl', 'Chandler', and 'Pink Popcorn' via ABP. (**a**) The expression profiles of ABP structural genes during three development stages in fruit of 'Black Pearl', 'Chandler', and 'Pink Popcorn' were analyzed using RNA-Seq. 'Black Pearl', 'Chandler', and 'Pink Popcorn' were abbreviated as H, Q, and F, respectively. (**b**) The expression profiles of structural genes in ABP were verified via qRT-PCR (mean ± SD from three biological replicates).

We analyzed the expression levels of these anthocyanin synthesis structural genes from Stage 1 to Stage 3 in 'Black Pearl', 'Chandler', and 'Pink Popcorn' fruits by using qRT-PCR to verify the results in RNA-Seq. The expression patterns of these genes were very similar to the results in RNA-Seq, and the detection results showed that these genes have consistent expression (Figure 4b; Supplementary Table S1).

### 3.3. Anthocyanin Biosynthesis-Related Transcription Factors (TFs) by WGCNA and Correlation Analysis

The expression levels of the structural genes in anthocyanin biosynthesis were regulated by the R2R3-MYB transcription factor [24]. All differentially expressed R2R3-MYB in the anthocyanin biosynthesis of blueberry were compared with *Arabidopsis thaliana* R2R3-MYB to construct a phylogenetic tree. Three R2R3-MYBs were highly homologous to the *Arabidopsis* R2R3-MYB (*AtMYB75/90/113/114*) that regulates anthocyanin biosynthesis (Figures 5a and S2). Based on phylogeny and expression levels, *VcMYBA* (maker-VaccDscaff1486-snap-gene-0.3) was identified as a potential TF candidate gene for regulating anthocyanin biosynthesis in blueberry fruit. *VcMYBA* was progressively up-regulated during the fruit coloring stages of 'Black Pearl', 'Chandler', and 'Pink Popcorn' (Figure 5b,c), similar to the expression trend of key structural genes for anthocyanin synthesis (Figure 4a).

To further confirm the potential function of *VcMYBA* in blueberry fruit coloration, we analyzed all DEGs using weighted gene co-expression network analysis (WGCNA). Thirty-five co-expressed gene modules were identified (Supplementary Figure S1), and *VcMYBA* co-expression with key structural genes, including *VcCHS3*, *VcDFR1*, *VcANS1*, *VcANS3*, *VcUFGT1*, and *VcUFGT2*, was determined in the dark green module (Figure 5d). The results of the phylogenetic tree, heatmap, and WGCNA analysis showed that the expression pattern of *VcMYBA* was positively correlated with that of anthocyanin synthesis structure genes during blueberry fruit coloring. *VcMYBA* is a candidate gene that may regulate the expression of anthocyanin synthesis as a structural gene in blueberries.

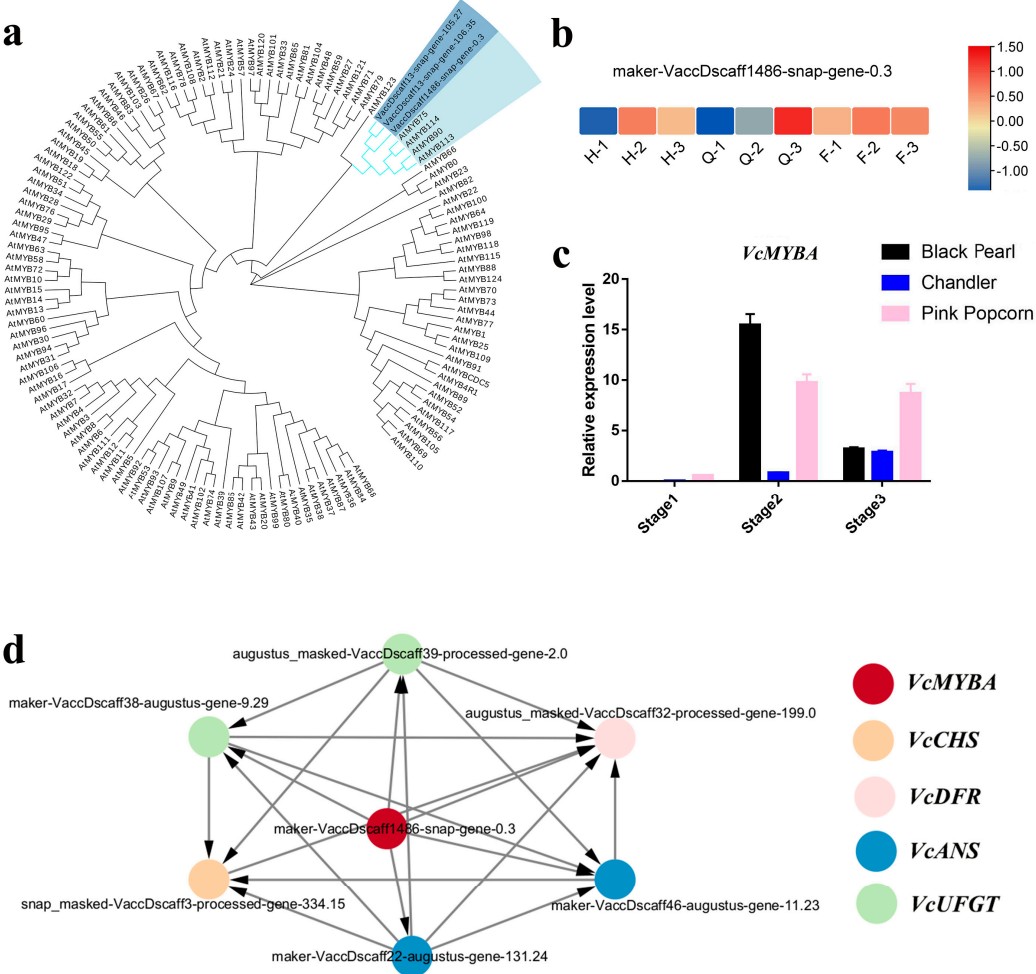

**Figure 5.** The candidate gene *R2R3-MYB* is involved in blueberry coloring. (**a**) Neighbor-joining (NJ) phylogenetic tree based on amino acid sequences of candidate R2R3-MYB in blueberries and all R2R3-MYBs of *Arabidopsis*; (**b**) illustration of its expression profile during blueberry fruit coloring by heatmap analysis based on the FPKM values of candidates *VcMYBA*; (**c**) the expression profiles of *VcMYBA* during fruit development using qRT-PCR. The data are the mean ± SD from three biological replicates; (**d**) the relationship between *VcMYBA* and ABP structural genes through WGCNA.

## 4. Discussion

### 4.1. Delphinidin Is the Main Anthocyanin Component for Mature Blueberry Fruit Color Difference between Black, Blue, and Pink

As secondary metabolites of plants, flavonoids play a crucial role in many biological processes and participate in the coloring of plant organs [62,63]. Variations in the colors of fruits are due to the presence of different anthocyanins [64]. *Vaccinium corymbosum* L. and *Vaccinium myrtillus* L. mainly contain delphinidin-3-O-galactoside [65]. Colorless fruits are mostly caused by the non-accumulation of anthocyanidin. Pink Popcorn is a rare variety of blueberry with pink-colored fruit.

To reveal the coloring mechanism of blueberries, anthocyanins from 'Black Pearl', 'Chandler', and 'Pink Popcorn' were phenotypically identified, and the LC-MS/MS metabolome was analyzed. The phenotypic identification results showed that the anthocyanin content in 'Black Pearl' and 'Chandler' gradually increased during the deepening of fruit color. During the fruit post-coloring process, the anthocyanin contents in 'Black Pearl' and 'Chandler' were always significantly higher than that in 'Pink Popcorn'. The pink color of 'Pink Popcorn' is caused by its extremely low anthocyanidin content (Figures 1 and 2). Delphinidin-, cyanidin-, malvidin-, peonidin-, and petunidin-based anthocyanins were detected. Addition-

ally, delphinidin-based anthocyanins with the highest content might be mainly responsible for the formation of their fruit color (Figure 3; Supplementary Table S2), which is consistent with the results of a previous study [41]. At Stage 3, various anthocyanin contents in 'Black Pearl' were significantly higher than that found in 'Chandler' and 'Pink Popcorn', which was consistent with its dark-black fruit color. Additionally, the reason for the pink color of 'Pink Popcorn' was that the anthocyanin content was low, being hard to detect in this study.

### 4.2. VcANS Is the Key Candidate Genes Responsible for Mature Blueberry Fruit Color

Generally, the changes in flavonoid biosynthetic metabolite content conform to changes in the expression level of structural genes in the corresponding biosynthetics [66,67]. In this study, the transcriptome and metabolome analyses were combined to analyze the fruit-coloring mechanisms in three blueberry varieties. A total of 18 structural genes were identified as candidate genes for anthocyanin synthesis in blueberries, including *VcCHS*, *VcCHI*, *VcF3H*, *VcF3′H*, *VcF3′5′H*, *VcDFR*, *VcANS*, *VcUFGT*, *VcANS*, and *VcLAR*. These genes showed different expression patterns during the three coloring stages of 'Black Pearl', 'Chandler', and 'Pink Popcorn'. Except for *VcF3′5′H*, there was no significant change found in the expression levels of other structural genes compared with the expression levels in 'Pink Popcorn'. Additionally, it was significantly lower in 'Pink Popcorn' than in the other two varieties after Stage 2.

*VcCHS* and *VcANS* were hardly expressed in the fruit of 'Pink Popcorn' during the coloring stage (Figure 4a), which is likely responsible for the pink fruit color of 'Pink Popcorn'. However, procyanidins, as downstream products of anthocyanins, exist in the fruit of 'Pink Popcorn' (Figure 3c). It means that *VcCHS*, as the first key enzyme in anthocyanidin synthesis, played a corresponding role in 'Pink Popcorn' despite its low expression. The transcriptome data showed a low expression level of *VcANR* in both 'Pink Popcorn' and 'Black Pearl' but a high expression level in 'Chandler', while two *VcLAR*s were highly expressed in 'Pink Popcorn'. *VcANR* may be the key candidate gene leading to higher procyanidin content in 'Chandler' than in 'Pink Popcorn' and 'Black Pearl'. Because of the low expression level of *VcANS* and *VcANR,* the leucocyanidin may be catalyzed by LAR enzymes to produce procyanidins in 'Pink Popcorn'. The encoding sequence of *ANS* has been identified, and the expression level of *ANS* is positively related to the content of anthocyanidin [67–69]. *VcANS* catalyze colorless leucocyanidin and leucodelphinidin into cyanidin and delphinidin, which is a key step in anthocyanin coloring [70–72]. The lack of *ANS* expression may result in reduced anthocyanin levels and weaker color [73–76]. The results are similar to those found in sweet oranges [77]. In summary, *VcANS* is the most likely candidate gene for the pink fruit color of 'Pink Popcorn'. The knock-out of the *VcANS* gene in blueberries could verify its function in other blueberry cultivars; we will see in the next step of this research.

### 4.3. VcMYBA Is a Key Candidate TFs Involved in Anthocyanin Synthesis in Vaccinium spp.

We identified a key transcription factor, *VcMYBA*, which has an up-regulated function during the coloring changes of three blueberry types, which was highly homologous to the notion that *MYB* regulates anthocyanin synthesis in *Arabidopsis thaliana* (Figure 5). The results showed that *VcMYBA* was co-expressed, with high expression levels of four anthocyanin synthesis key structural genes (*VcCHS*, *VcDFR*, *VcANS*, and *VcUFGT*) in blueberry fruit. R2R3-MYB is an important transcription factor regulating structural genes related to ABP [32,78,79]. *VcMYBA*, as a part of the MBW model, can strongly activate *DFR* and participate in the regulation of anthocyanin biosynthesis in blueberries [17,54–57]. *VcMYBA* may affect blueberry fruit coloration by activating the expression of structural genes, which should be further investigated as an important candidate gene to advance fruit color breeding in blueberry varieties.

## 5. Conclusions

Blueberries have a rich fruit color determined by their anthocyanidin content. Typically, the fruit of 'Black Pearl' and 'Chandler' are black to blue, but the fruit of 'Pink Popcorn' are pink in color. The phenotypic characteristics of the three blueberry varieties were verified via pigment identification, quantitative analysis, and metabolome analysis of anthocyanin levels. A higher content of delphinidins and cyanidins may result in darker fruit color, as in the 'Black Pearl' variety, but in 'Pink Popcorn', these anthocyanins levels were nearly zero. The structural genes involved in ABP were identified via transcriptome analysis, in which *VcANS* was found to be the key candidate gene related to the pink fruit color of 'Pink Popcorn'. At the same time, a candidate transcription factor, *VcMYBA*, was found to activate the expression of four anthocyanin biosynthesis structural genes related to the coloring of blueberries. The findings in this study have enriched our understanding of color formation in blueberries. The above results can be applied for further breeding work on blueberry fruit color through genomic editing.

**Supplementary Materials:** The following supporting information can be downloaded at: https://www.mdpi.com/article/10.3390/horticulturae10010105/s1, Figure S1: Weighted gene co-expression network analysis (WGCNA) during fruit coloring stages in three blueberry varieties. (a) Hierarchical clustering tree of unigenes based on their expression level in 27 transcriptomes. Each branch represents an unigene, and each color below represents a module. The dynamic tree cut shows that the unigenes fall into different modules. The merged dynamic indicates that the modules are divided by clustering modules with similar expression patterns; (b) the heatmap analysis displays the module expression pattern in the 27 samples. 'Black Pearl', 'Chandler', and 'Pink Popcorn' are abbreviated as H, Q, and F, respectively. Each stage of sampling H, Q, and F had three biological replicates; Figure S2: Alignment of the amino acid sequence among VaccDscaff1486-snap-gene-0.3 (VcMYBA), AtMYB75, AtMYB90, AtMYB113, and AtMYB114; Table S1: Primers used in this study; Table S2: The content of anthocyanin compounds in fruit of 'Black Pearl', 'Chandler', and 'Pink Popcorn'; Table S3: DEGs, TF, and reference genes related to flavonoid and ABP.

**Author Contributions:** Conceptualization, G.X. and L.C.; methodology, L.C.; software, Q.D.; validation, H.W. and G.X.; formal analysis, Q.D.; investigation, A.L., H.L. and Z.L.; resources, H.W.; data curation, G.L. and Y.D.; writing—original draft preparation, Q.D.; writing—review and editing, L.C and G.X.; visualization, Q.C. and Y.L.; supervision, H.W. and G.X.; funding acquisition, G.X and L.C. All authors have read and agreed to the published version of the manuscript.

**Funding:** This research was funded by [The Department of Science and Technology of Liaoning Province] grant number [2022020655-JH1/109] and [The Dalian Science and Technology Bureau] grant number [2022RQ068]; The APC was funded by [Guohui Xu].

**Data Availability Statement:** The data that support the findings of this study are available from the corresponding author upon reasonable request. The data is private due to the use in other unpublished articles.

**Conflicts of Interest:** The authors declare no conflicts of interest.

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
